# Virus-Induced Gene Silencing (VIGS) in Chinese Jujube

**DOI:** 10.3390/plants12112115

**Published:** 2023-05-26

**Authors:** Yao Zhang, Nazi Niu, Shijia Li, Yin Liu, Chaoling Xue, Huibin Wang, Mengjun Liu, Jin Zhao

**Affiliations:** 1College of Life Science, Hebei Agricultural University, Baoding 071000, China; zhangyao02091221@126.com (Y.Z.); niunazi980219@163.com (N.N.); lishijia1125@126.com (S.L.); 13102937376@163.com (Y.L.); xuechaoling2023@126.com (C.X.); wanghuibin069@163.com (H.W.); 2Research Center of Chinese Jujube, Hebei Agricultural University, Baoding 071000, China; lmj1234567@aliyum.com

**Keywords:** virus-induced gene silencing, tobacco rattle virus, Chinese jujube, pTRV2-ZjCLA, twice injections, gene function

## Abstract

Virus-induced gene silencing (VIGS) is a fast and efficient method for assaying gene function in plants. At present, the VIGS system mediated by Tobacco rattle virus (TRV) has been successfully practiced in some species such as cotton and tomato. However, little research of VIGS systems has been reported in woody plants, nor in Chinese jujube. In this study, the TRV-VIGS system of jujube was firstly investigated. The jujube seedlings were grown in a greenhouse with a 16 h light/8 h dark cycle at 23 °C. After the cotyledon was fully unfolded, *Agrobacterium* mixture containing pTRV1 and pTRV2-ZjCLA with OD_600_ = 1.5 was injected into cotyledon. After 15 days, the new leaves of jujube seedlings showed obvious photo-bleaching symptoms and significantly decreased expression of *ZjCLA*, indicating that the TRV-VIGS system had successfully functioned on jujube. Moreover, it found that two injections on jujube cotyledon could induce higher silencing efficiency than once injection. A similar silencing effect was then also verified in another gene, *ZjPDS*. These results indicate that the TRV-VIGS system in Chinese jujube has been successfully established and can be applied to evaluate gene function, providing a breakthrough in gene function verification methods.

## 1. Introduction

With the development of molecular biotechnology, the whole genome sequencing of many plants has been completed in recent years [1], and the gene function verification has become more and more important. Traditional gene function identification methods include antisense transgenic technology and gene knockout technology [2], but these methods are complex and time-consuming. VIGS technology, meanwhile, has become a powerful tool for gene function identification due to its fast, efficient, and high-throughput characteristics [3]. Gene silencing is divided into transcriptional gene silencing (TGS) at the transcription level and post-transcriptional gene silencing (PTGS) [4,5]. TGS occurs in the nucleus, and gene transcription cannot be carried out normally due to promoter inactivation, while PTGS refers to the normal transcription of genes, but the accumulation of transcribed mRNA in the cytoplasm is very low or else not detectable at all [6]. Virus-induced gene silencing is post-transcriptional gene silencing, which can activate the plant immune system by infecting the host plant with the virus vector containing the endogenous gene fragments [7], and while identifying and degrading viral RNA, microRNA containing endogenous target genes are also generated [8]. These microRNAs bind to the mRNA of target genes and are then degraded by Dicer enzyme, thus reducing the expression level of target genes or losing their function [9]. In this system, phenotypic mutations occurred mainly through gene silencing induced by viruses carrying functional cDNA after infecting plants [10], and the gene function can be identified by changes in plant phenotype or physiological indicators [11].

VIGS vectors can be divided into RNA, DNA, and satellite virus vectors. Common RNA virus vectors include Tobacco mosaic virus (TMV) [12], Potato X virus (PXV) [13], Tobacco rattle virus (TRV) [14], Cotton leaf crumple virus (CLCrV) [15], Bean pod mottle virus (BPMV) [16], Barley stripe mosaic virus (BSMV) [17], and so on. Among them, TRV-induced gene silencing (TRV-VIGS) has become the most widely used system due to many advantages, such as high silencing efficiency, long duration, mild virus symptoms in host plants, and gene silencing in various tissues [18]. To date, TRV-VIGS has been successfully applied to *Arabidopsis*, tobacco, rice, corn, tomato, and strawberry fruit [14,19,20,21].

In order to study the silencing effect of the VIGS carrier, commonly used marker genes include chloroplastos alterados 1 gene (CLA1), phytoene desaturase (PDS), green fluorescent protein (GFP), and so on [22,23,24]. Among them, *CLA1* encodes the 1-deoxy-D-xylulose 5-phosphate synthase, the first enzyme of the 2-C-methyl-d-erythritol-4-phosphate pathway associated with chloroplast development, which is highly conserved in a variety of plants [24]. In cotton, the albino mutation of cotton was obtained by silencing *CLA1* gene [19]. In strawberry, TRV-mediated VIGS technique was used to construct a silencing system with PDS and GFP as marker genes [25]. The leaves and fruits of silenced plants showed an obvious bleaching phenomenon, and the expression of *PDS* in silenced plants was significantly reduced. The GFP silenced plants can be easily screened by fluorescence microscope and ultraviolet lamp. Anthocyanins, a kind of flavonoid, form gradually in the late ripening period of fruit and have an important effect on skin color. In apple, VIGS technology was used to silence key genes in flavonoid-synthesis pathways, such as anthocyanin synthase, chalkeone isomerase, dihydroflavonol reductase, and phenylalanine aminlyase, in order to identify key genes regulating anthocyanin synthesis, thereby providing theoretical guidance for molecular breeding [26].

Chinese jujube (*Ziziphus jujuba* Mill.) is native to China, and belongs to the Rhamnaleae family. Due to its high nutritional value, drought, and barren tolerances, jujube has multiple economic, ecological, and social benefits. In recent years, the research on molecular biology and genetic engineering of important biological traits of jujube has been deepened gradually, and more and more functional genes have been excavated and identified [27,28]. As the information on the genomic organization accumulates, gene functional analysis must be accelerated in jujube. As a perennial woody plant, it has been reported that exogenous genes were transferred into the stem tip, leaf, and hypocotyl of jujube by the traditional *Agrobacterium* mediated method [29]. However, due to the time consuming, low transferred efficiency and unstable technology, it is still difficult to extensively apply it in gene function studies on jujube. Thus, it is urgent to establish a fast and efficient gene function verification system.

To build an efficient VIGS system, many factors should be considered, among which the selection of the plant virus will affect the efficiency of VIGS. Different viruses have different characteristics, and host range, infection method, ambient temperature of plant growth after infection [30], size of inserted fragments [31], inoculation concentration, and plant age will all affect the silencing efficiency. Therefore, different plants have their own optimal infection system, and it is necessary to construct the optimal VIGS system for the target species [32].

In this study, we aim to establish a VIGS system based on the TRV virus in Chinese jujube. First, *ZjCLA*, an easily observable marker gene, was selected. After silencing *ZjCLA* in jujube, the feasibility of the VIGS-TRV system was evaluated by phenotypic observation and quantitative detection. The system was then optimized with a second injection, where the silencing efficiency was obviously improved. Using the optimized system, a similar result was also verified by another marker gene, *ZjPDS*. Thus, an easily operated, high-efficiency VIGS-TRV system was established, providing technical support for exploring the gene function in jujube.

## 2. Results

### 2.1. TRV Can Infect Jujube Seedlings

To demonstrate whether Chinese jujube can be infected with tobacco rattle virus (TRV), *Agrobacterium* solution containing pTRV1 and pTRV2 was injected into the cotyledon of jujube seedlings. As shown in Figure 1A, healthy jujube seedlings with fully expanded cotyledon were selected for injection. Compared with the control plants, the cotyledon injected with the *Agrobacterium* solution could obviously be damp. At 15 d after injection, the seedlings injected with the virus-bacteria solution survived normally and had no significant morphological differences compared with the controls (Figure 1A). To further verify the infection efficiency, the presence of TRV1 and TRV2 transcripts in the seedlings was confirmed using the specific primers, and the fragments were sequenced (Figure 1B). These results indicated that recombinant TRV can be efficiently replicated and transmitted in jujube seedlings.

### 2.2. The Construction of pTRV2-ZjCLA Recombination

*CLA* gene is usually used to evaluate VIGS systems due to leaves with *CLA*-silenced plants being easily recognizable by photobleaching symptoms, and, thus, *ZjCLA* was selected and cloned as a target gene in this study. To achieve efficient gene silencing, it is necessary to select some fragments that are able to form efficient siRNAs, thereby reducing the off-target rate [33]. A 210-bp fragment with 42% GC content in *ZjCLA* was selected for constructing the recombination with pTRV2 based on predicted siRNA target sequences (Figure 2A,B). At 15 d after injection, the target fragment was detected successfully in newly leaves of jujube by PCR (Figure 2C), suggesting that *Agrobacteria* carrying a pTRV2-*ZjCLA* recombination were transferred on jujube with the assistance of pTRV1.

### 2.3. Silencing of ZjCLA in Chinese jujube

To test whether the TRV-based vector can effectively induce the silencing of *ZjCLA* in jujube, the phenotypes and the expressions of *ZjCLA* of injected jujube seedlings were investigated. After 15 days post-inoculation (dpi), the photobleaching phenotype was observed in all real leaves of jujube seedlings infected with pTRV1+2-*ZjCLA* (Figure 3A). Meanwhile, under the same growth conditions, the control seedlings infected by pTRV1+2 showed developed normal leaves without photobleaching symptoms (Figure 3B). A total of 65% of 60 jujube seedlings infected with pTRV1+2-*ZjCLA* exhibited a large area of photo-bleaching or chlorosis phenotype in their new leaves.

To further verify the silencing efficiency of *ZjCLA*, the mRNA level of *ZjCLA* in the real leaves of jujube was measured by using qRT-PCR (Figure 3B). Compared with the controls, the transcripts of *ZjCLA* were significantly decreased in leaves of jujube inoculated with pTRV1+2-*ZjCLA* (Figure 3C). The contents of chlorophyll in the silenced and control leaves were then further detected. The chlorophyll content in silenced leaves was more significantly reduced than the controls (Figure 3D). These results showed that *ZjCLA* was specifically silenced in pTRV1+2-*ZjCLA* infected seedlings, indicating that pTRV-VIGS system was successfully established in jujube.

### 2.4. Optimization of the TRV-VIGS System in Jujube

In order to improve the efficiency of silencing, the second injection was applied after 7 d of the first injection. The same concentration of pTRV1 and pTRV1+2-*ZjCLA Agrobacterium* solution was used in the second injection. After 15 dpi, the expression level of *ZjCLA* in the leaves of jujube seedlings with one and two injections were detected. The results showed that the expression of *ZjCLA* in jujube seedlings with two injections was significantly lower than that in seedlings with one injection (Figure 4A). In addition, the seedlings with one injection showed a recovery phenotype after 30 dpi, whereas the seedlings with two injections could keep bleaching phenotype at 50 dpi (Figure 4B). The results showed that two injections could significantly improve the silencing efficiency and prolong the silencing period.

### 2.5. ZjPDS Silencing by Using the TRV-VIGS System in Jujube

To confirm whether the TRV-VIGS system is suitable for other genes in jujube, another marker gene, *ZjPDS*, was further applied. PDS encodes an enzyme catalyzing the first step in the carotenoid biosynthetic pathway, and the knockdown of its transcript will result in photobleaching leaves. One fragment of 207 base pairs of *ZjPDS* containing the predicted siRNA target sequences was identified (Figure 5A). The *Agrobacterium* mixture containing pTRV2-*ZjPDS* and pTRV1 was infiltrated into the cotyledon of jujube seedlings twice. After 15 dpi, the newly grown leaves of jujube seedlings inoculated with pTRV1+2-*ZjPDS* were turning yellow, and the control seedlings displayed normal leaves without symptoms (Figure 5B). In the meantime, qRT-PCR analysis showed that the expressions of *ZjPDS* were significantly down-regulated in the silencing seedlings (Figure 5C). The results further confirmed that the established TRV-VIGS system has general applicability for the genes in jujube.

## 3. Discussion

In previous studies, the gene function in plants was mainly identified through stable genetic transformation and mutant phenotype analysis [34]. However, for some non-model plants, especially fruit trees, it is difficult to study gene functions through traditional genetic transformation methods. Chinese jujube is a characteristic fruit tree, the genomes of which have been sequenced [27,35,36]. Therefore, the functional analysis of key genes can contribute to improving the quality and biological characteristics of jujube through molecular breeding. However, as a woody plant, jujube has a long growth cycle, high heterozygosity, and a low genetic transformation rate [37], which greatly hinders gene function research in jujube. Thus, developing an easy and feasible approach for the evaluation gene function is of great significance in jujube.

VIGS technology has been considered one of the quickest approaches to evaluate the gene function. Among nearly 40 kinds of viruses, TRV is the most widely used virus to silence genes in plants so far [1,38]. It is applicable to many plants because of its advantages, such as broad host range, highly efficiency, long persistence, and the fact that it is harmless to the host plants [25,39,40]. In peach, silencing CHS genes could reduce the anthocyanin content of fruit peel and regulate anthocyanin metabolism pathways, further demonstrating that CHS can control the metabolic direction of flavonoids [41,42]. In kiwifruit, after AcTPR2 was silenced, defense enzymes such as SOD and POD and endogenous hormones such as IAA and GA_3_ were successfully activated to cope with the infection of *Botrytis cinerea* [43]. However, the TRV vector is not suitable for all plants. In papaya, the TRV-VIGS system could not induce the corresponding phenotype when the endogenous gene was silenced [44].

In this study, the efficiency of TRV in silencing the *CLA* and *PDS* genes through *Agrobacterium* infection were verified in jujube. After 15 dpi, the photobleaching phenotype in newly grown leaves was clearly observed and the expressions of *ZjCLA* and *ZjPDS* were obviously decreased. The results showed that the TRV-VIGS system can be used for rapid gene function analysis in jujube. Next, TRV-GFP can be constructed to determine whether they are silenced by fluorescence observation phenotype. Overall, this work provides a visualizable and efficient method to achieve gene silencing in jujube.

The silencing efficiency about VIGS-TRV system depends on the dynamic interaction between virus propagation and plant growth. Some studies have been shown that the VIGS efficiency gradually declined with the change of plant development because the virus could not infect organs formed at later development [18,45]. In this study, two injections can significantly increase silencing efficiency, providing useful clues for other VIGS studies, especially in wood plants.

## 4. Materials and Methods

### 4.1. Cultivation of Jujube Seedlings

For assuring the source consistency, healthy seeds of the same jujube tree (*Ziziphus jujuba* var. spinosa (Bunge) Hu) were used and rinsed with fresh water for 3–5 times. After soaking overnight, the surface water of the seeds was absorbed and then wrapped with two layers of gauze and placed in a petri dish. Each petri dish placed 30 seeds in a constant temperature incubator for 2–3 d (28 °C and 16 h light/8 h darkness). Water is sprayed periodically to keep the gauze moist during culture. When the radicles of jujube seeds sprouted to about 1 cm, they were transplanted into soil and cultured for a period of time until the cotyledon was fully developed for infection.

### 4.2. Identification and Cloning of ZjCLA and ZjPDS in Jujube

The *CLA* and *PDS* genes sequence was downloaded from the jujube genome database, and the sequence characteristics were analyzed by ClustalX and DNAMAN to verify their accuracy. Through the SGN-VIGS website (https://vigs.solgenomics.net/, (accessed on 20 November 2021)), the silent efficient DNA sequences were predicted [46]. A specific fragment about 200 bp in length was then selected in the ORF frame. Primer Premier 5.0 was used to design primers according to the principles of in-fusion primer design. The primers were listed in Table 1. PCR products were purified and cloned into the pMD 19 vector (TIANGEN, Beijing, China) and sequenced.

### 4.3. pTRV2-ZjCLA and pTRV2-ZjPDS Vector Construction

The pTRV2 plasmid was double-digested with the restriction enzymes FastDigest *Bam*HI/*EcoR*I (FD0054/FD0274) and FastDigest *Xho*I (FD0694) (Thermo, Waltham, MA, USA). The reaction system consisted of pTRV2 plasmid 20 μg, *Bam*HI/*EcoR*I 2 μL, *Xho*I 2 μL, Buffer 6 μL, and ddH_2_O supplementation to 50 μL, and the enzyme was digested at 37 °C for 30 min. In order to construct the pTRV2-*ZjCLA (*pTRV2-*ZjPDS)* vector, the specific fragment of *ZjCLA* (*ZjPDS*), which had *EcoR*I/*Xho*I (*Bam*HI/*Xho*I) restriction sites, was amplified. The PCR product was then linked to the digested pTRV2 plasmid using pEASY-Uni Seamless Cloning and Assembly Kit (TransGen Biotech, Beijing, China), and transformed into *Escherichia coli* (DH5α). The positive clones were identified by PCR. After PCR detection, the bacterial solution containing the target bands was sent to Tsingke (Beijing, China) for sequencing. The primers are listed in Table 1.

### 4.4. Preparation and Infection of Bacterial Solution

Recombinant plasmid pTRV2-*ZjCLA* and pTRV2-*ZjPDS* with correct sequencing was taken to transform *Agrobacterium* receptive GV3101. For 2 d culture at 28 °C, single rounded plaque was selected and cultured in 1 mL liquid LB medium containing 50 μg/mL kanamycin and 50 μg/mL rifampicin at 28 °C. The shaken bacterial solution was transferred into the 50 mL LB medium, which contained 25 μL rifampicin (50 mg/mL), 25 μL kanamycin (100 mg/mL), 2.5 mL MES (pH 5.7, 0.2 M), and 10 μL acetosyringone (0.1 M) at a ratio of 1:20 for amplification culture, and the culture was incubated at 28 °C for 12–16 h. The cultured induced LB bacteria solution was centrifuged at 6000 rpm for 20 min, and then the supernatant was abandoned and the bacteria were re-suspended with the infiltration buffer (10 mM MES, 10 mM MgCl_2_ and 200 μM AS). The final value of OD_600_ was adjusted to 1.5, and stood at room temperature for 2–3 h. For injection, the bacteria solution containing pTRV1 and bacteria solution containing pTRV2-*ZjCLA* or pTRV2-*ZjPDS* were mixed at a volume ratio of 1:1.

### 4.5. Infection of Jujube Seedlings

The jujube seedlings were injected with a syringe. Using a needle to make a slight scar on the back of the cotyledon, a 1 mL sterile syringe without the needle was then used to align the wound and inject the mixed bacteria solution. The infected jujube seedlings were first dark-treated for 24 h, and then cultured in a culture chamber (light 16 h/dark 8 h, 23 °C, 60% of humidity).

### 4.6. RNA Isolation and Gene Expression Analysis

Total RNA was extracted using an RNA Easy Fast Plant Tissue Kit (TIANGEN, Beijing, China). After genomic DNA was removed by RNase-free DNase I (TIANGEN, Beijing, China), RNA concentrations and purities were checked with a NanoDrop2000 Spectrophotometer (Thermo Scientific). First strand cDNA was then synthesized by reverse transcription of 500 ng of total RNA with a FastQuant RT Super Mix Kit (TIANGEN, Beijing, China).

Total RNA was extracted from the 2–6 true leaves of jujube and the total RNA was obtained by reverse transcription to cDNA. qRT-PCR primers were designed according to the CDS sequences of *ZjCLA* and *ZjPDS*. The PCR efficiencies of the two-pair primers were 99.0 to 103.0%, respectively. A single PCR amplification band of the expected size for *ZjCLA*/*ZjPDS* was detected, and the melting curves of these primers also showed a single peak. Above detections indicated that these primers were efficient and specific. *ZjACT* was used as internal reference [47]. The silencing efficiency of the seedlings injected by pTRV2-*ZjCLA*/*ZjPDS* was calculated. The primers are listed in Table 1.

### 4.7. Determinations of Chlorophyll Content

Plant chlorophyll content ELISA Kits (MEIMIAN, Yancheng, China) were used to determine chlorophyll content. According to the instructions, approximately 0.1 g of jujube leaves was ground in liquid nitrogen and transferred to a 2 mL tube containing 900 μL of PBS (pH 7.2) for extraction followed by centrifugation at 2000–3000 rpm for 20 min, and the supernatant was retained for detection. Chlorophyll content was then detected on a Tecan Infinite M200 PRO full-wave length multifunctional enzyme labeling instrument (Tecan, Hombrechtikon, Switzerland). At least three biological replicates were performed in each treatment.

### 4.8. Statistical Analyses

The data were analyzed with Excel 2016 software (Microsoft Corp., Redmond, WA, USA). All statistical analyses were performed with SPSS software 17.0. Duncan’s multiple range tests were used to assess differences between different treatments. Different letters mean significant a difference at level of *p* < 0.05 between the corresponding treatments.

## 5. Conclusions

In this study, a TRV-VIGS system in Chinese jujube was established. The jujube seedlings were grown in a greenhouse with a 16 h light/8 h dark cycle at 23 °C. After the cotyledon was fully unfolded, *Agrobacterium* mixture containing pTRV1 and pTRV2-*ZjCLA* or pTRV2-*ZjPDS* with OD_600_ = 1.5 was injected into cotyledon. After 15 dpi, the target gene in new leaves of seedlings was successfully silenced. Moreover, two injections can significantly increase silencing efficiency and keep the silencing effect at least 50 d. The successful silencing *ZjCLA* and *ZjPDS* indicates that the optimized TRV-VIGS system was established in jujube, which is of great significance for its functional genomics.

## Figures and Tables

**Figure 1 plants-12-02115-f001:**
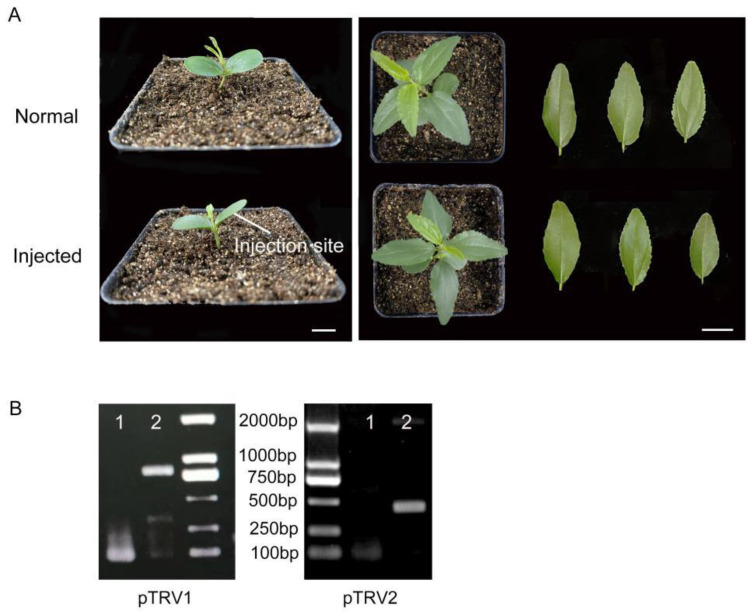
TRV can infect jujube seedlings. (**A**) Jujube seedlings with fully expanded cotyledon were selected for TRV treatments, and the back of the cotyledon was selected as the injection site. (**B**) Detection of TRV fragments after VIGS treatments. 1, Normal leaves; 2, Injected leaves. The bars represent 1 cm.

**Figure 2 plants-12-02115-f002:**
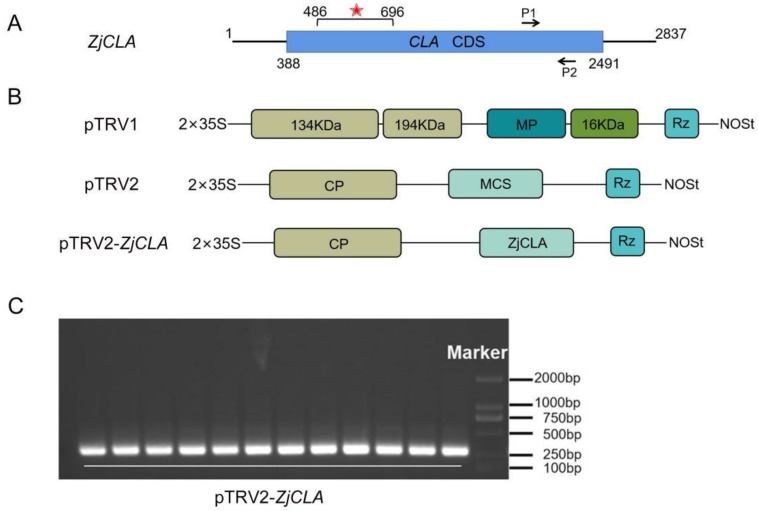
pTRV2-*ZjCLA* construction and detection in jujube. (**A**) The bold line represents the mRNA of *ZjCLA*, the blue box represents the open reading frame of the gene, the number represents the corresponding base position on the mRNA, the silent fragment surrounded by VIGS illustrated by the red asterisk, and the arrow (P1 and P2) represents the binding site of the qRT-PCR primer on each fragment. MP (movement protein), CP (coat protein), MCS (multiple cloning sites), Rz (self-cleaving ribozyme), NOSt (nopaline synthase terminator). (**B**) Genomic organization of the TRV components, pTRV1, pTRV2 and pTRV2-*ZjCLA*. (**C**) The target fragments of *ZjCLA* were detected in jujube seedlings after injection. The denaturizing temperature of each sample hole was different, and the temperatures ranged from 55–65 °C.

**Figure 3 plants-12-02115-f003:**
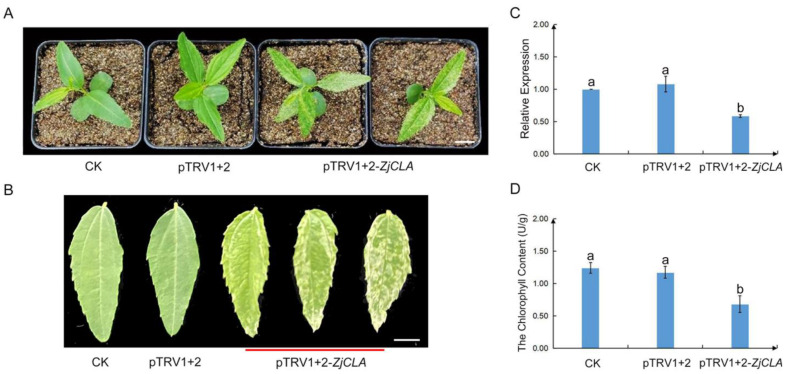
Silencing *ZjCLA* in jujube seedlings using the pTRV1+2-*ZjCLA*. (**A**) Jujube seedlings under different treatments. CK represents untreated, normal growth of jujube seedlings, pTRV1+2 represents the jujube seedlings inoculated with pTRV1 + pTRV2, pTRV1+2-*ZjCLA* represents the jujube seedlings inoculated with pTRV1 + pTRV2-*ZjCLA*. Bar = 1 cm. (**B**) The leaves of CK, pTRV1+2 and pTRV1+2-*ZjCLA*. (**C**) The expression of *ZjCLA* in leaves of CK, pTRV1+2, and pTRV1+2-*ZjCLA* by qRT-PCR analysis. (**D**) The chlorophyll content in leaves of CK, pTRV1+2, and pTRV1+2-*ZjCLA*. At least three replicates were used in each treatment. All statistical analyses were performed with SPSS software 17.0. Different letters above the columns indicate significant differences between treatments according to Duncan’s multiple range test (*p* < 0.05).

**Figure 4 plants-12-02115-f004:**
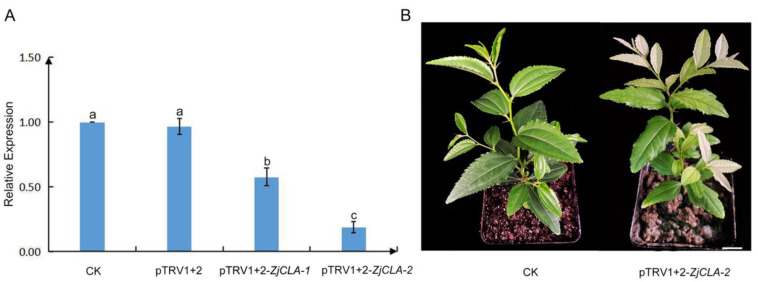
The comparison of silenced effect between once and twice injected pTRV1+2-*ZjCLA*. (**A**) The expressions of *ZjCLA* in leaves of jujube seedlings after one and two injections. CK represents untreated, normal growth of seedlings, pTRV1+2 mean jujube seedlings injected by pTRV1 + pTRV2. pTRV1+2-*ZjCLA*-1 and pTRV1+2-*ZjCLA*-2 mean one and two injections of pTRV1 + pTRV2-*ZjCLA*, respectively. At least three replicates were used in each treatment. All statistical analyses were performed with SPSS software 17.0. Different letters above the columns indicate significant differences between treatments according to Duncan’s multiple range test (*p* < 0.05). (**B**) Phenotype of jujube seedlings at 50 days after two injections. Bar = 1 cm.

**Figure 5 plants-12-02115-f005:**
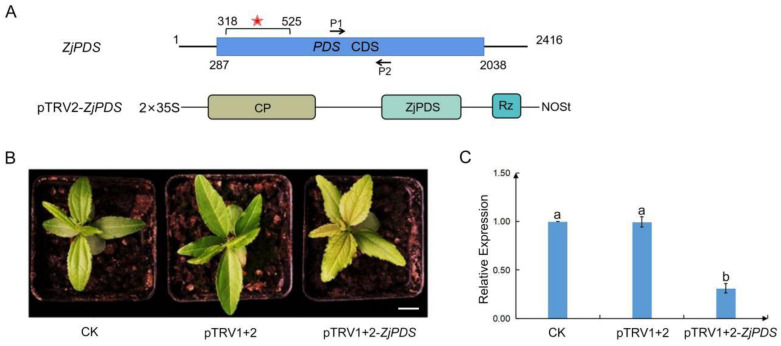
Silencing *ZjPDS* in jujube seedlings using the pTRV1+2-*ZjPDS*. (**A**) The bold line represents the mRNA of *ZjPDS*, the blue box represents the open reading frame of the gene, the number represents the corresponding base position on the mRNA, the silent fragment surrounded by VIGS is illustrated by the red asterisk, and the arrows (P1 and P2) represent the binding site of the qRT-PCR primer on each gene fragment. Genomic organization of the pTRV2-*ZjPDS.* CP (coat protein), Rz (self-cleaving ribozyme), NOSt (nopaline synthase terminator). (**B**) Jujube seedlings under different treatments. CK represents untreated, normal growth of seedlings, pTRV1+2 represents the seedlings inoculated with pTRV1 + pTRV2, pTRV1+2-*ZjPDS* represents the seedlings inoculated with pTRV1 + pTRV2-*ZjPDS*. Bar = 1 cm. (**C**) The expressions of *ZjPDS* in leaves of CK, pTRV1+2 and pTRV1+2-*ZjPDS* by qRT-PCR. At least three replicates were used in each treatment. All statistical analyses were performed with SPSS software 17.0. Different letters above the columns indicate significant differences between treatments according to Duncan’s multiple range test (*p* < 0.05).

**Table 1 plants-12-02115-t001:** The primers used in this study.

Usage	Primer Sequence
Partial gene cloning of *ZjCLA*	F: GAGCAGATCTATTGGGCTTAGC; R: ATGACATCAGACCTTAGTTCATCC
Vector construction of pTRV2-ZjCLA	F: TGTGAGTAAGGTTACCGAATTCGAGCAGATCTATTGGGCTTAGC R: GGGACATGCCCGGGCCTCGAGATGACATCAGACCTTAGTTCATCC
qRT-PCR of *ZjCLA*	F: ATCAATTGGAGGCTTTGGAT; R: ACTGTTGCTGCAATATGAGA
Partial gene cloning of *ZjPDS*	F: ACTTGAGCTGGCAAACTA; R: CACACAAACTACCTTCAAA
Vector construction of pTRV2-*ZjPDS*	F: AGAAGGCCTCCATGGGGATCCACTTGAGCTGGCAAACTA R: GGGACATGCCCGGGCCTCGAGCACACAAACTACCTTCAAA
qRT-PCR of *ZjPDS*	F: AGGAGAGTTCAGTCGATTTG; R: TGACATGGCAATAAACACCT
TRV virus testing of pTRV1	F: GCTCTTGGGAACTACATGGTG; R: CGCCTCAATCGTCTTCATCTCC
TRV virus testing of pTRV2	F: GGTACGTAGTAGAGTCCCAC; R: ACAAAAGACTTACCGATCAATC
qRT-PCR of *ZjACT*	F: TTGCTTCTCACCCTTGATGC; R: AGCCTTCCTGCCAACGAGT

## Data Availability

Data will be made available upon request.

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
