# Peer review of "Virus-Induced Gene Silencing (VIGS) in Chinese Jujube"

_plants, 2023, doi:10.3390/plants12112115_

Round 1

Reviewer 1 Report

While the results of this manuscript were clearly demonstrated, there needs to be a more thorough descriptions in the Materials and Methods section. As just one example, the authors use qRT-PCR to measure the relative decrease in expression levels of the silenced gene of interest but do not give any details on the primer efficiencies of the various primers utilized including the internal control ZjACT gene. You can not accurately quantify relative levels unless the primer efficiencies are nearly identical or the software used for the qPCR instrument has the ability to correct for differences in primer efficiencies. Was the ZjACT gene validated for steady expression throughout the various sampling times/conditions as is required for a good internal standard? The visual evidence of silencing was unmistakable but the quantification of the silencing needs to be tightened up.

Other points: 1) the authors should spell out the scientific name of jujube  (Ziziphus jujuba) in the Introduction to give some context to the naming of the various target genes. Also, what variety of jujube was used (Mill.?)?

2) Why test for the presence of the TRV DNA constructs as a way to demonstrate successful infection of the plants? It would be more accurate if the viral RNA levels were assessed as that is a truer reflection of the infection process since TRV is an RNA virus, not a DNA virus. Given that the DNA was PCR amplified, there is no mention that the resultant fragments were ever sequenced to see if it truly was TRV specific. Just because you detect a PCR fragment of the expected size does not necessarily demonstrate that the primers were specific for the targeted regions.

3) If the double injection for ZjCLA proved to be superior in terms of silencing, why wasn't the same done for the ZjPDS?

4) The scientific units need to be standardized. Is it uL or micro symbol L as both are used interchangeably? Also, the authors need to state the unit concentration of the various enzymes utilized. Merely stating the volume used is insufficient for reproducibility.

The English is a bit rough and needs to be improved in terms of proper sentence structure, vocabulary used, etc.

Author Response

Dear Reviewers,

Thank you for the comments concerning our manuscript entitled “Virus-induced gene silencing (VIGS) in Chinese jujube” ( plants-2405132 ). Those comments are very helpful for revising and improving our manuscript. We have studied your comments carefully and have made correction which we hope meet with approval. Revised portion are marked in yellow in the revised manuscript. Below you will find our detailed replies to the specific comments. Comments are pasted below in black text and our replies are in blue text. 

Reviewers’ comments are list as C.

Authors’ replies to the comments are list as R. 

To Reviewer 1:

C1: As just one example, the authors use qRT-PCR to measure the relative decrease in expression levels of the silenced gene of interest but do not give any details on the primer efficiencies of the various primers utilized including the internal control ZjACT gene. You cannot accurately quantify relative levels unless the primer efficiencies are nearly identical or the software used for the qPCR instrument has the ability to correct for differences in primer efficiencies. Was the ZjACT gene validated for steady expression throughout the various sampling times/conditions as is required for a good internal standard? The visual evidence of silencing was unmistakable but the quantification of the silencing needs to be tightened up.

R1: Thanks for your comment. In our previous study, ZjACT was screened out as an ideal internal reference gene (Bu et al. 2016), and which was cited in the revised manuscript.

Bu, J.; Zhao, J.; Liu, M. Expression stabilities of candidate reference genes for RT-qPCR in Chinese jujube (Ziziphus jujuba Mill.) under a variety of conditions. PLoS One. 2016, 11(4), e0154212.

C2: the authors should spell out the scientific name of jujube (Ziziphus jujuba) in the Introduction to give some context to the naming of the various target genes. Also, what variety of jujube was used (Mill.?)?

R2: Thanks for your comment. We have added the scientific name of jujube (Ziziphus jujuba) in the Introduction. ‘Mill.’ is the name abbreviation of the namer “Miller”.  Wild jujube (Ziziphus jujuba var. spinosa (Bunge) Hu) was used in this study, which was supplemented in the Materials and Methods.

C3: Why test for the presence of the TRV DNA constructs as a way to demonstrate successful infection of the plants? It would be more accurate if the viral RNA levels were assessed as that is a truer reflection of the infection process since TRV is an RNA virus, not a DNA virus. Given that the DNA was PCR amplified, there is no mention that the resultant fragments were ever sequenced to see if it truly was TRV specific. Just because you detect a PCR fragment of the expected size does not necessarily demonstrate that the primers were specific for the targeted regions.

R3:We are so sorry to make you confused. The viral RNA levels of infected jujube seedlings were detected by the specific primers of pTRV1 and pTRV2, and the fragments were also sequenced and verified. We rewrote the related content in the manuscript.

C4: If the double injection for ZjCLA proved to be superior in terms of silencing, why wasn't the same done for the ZjPDS?

R4: The double injection was used in silencing ZjPDS, and the similar silenced effect was also verified. In the revised manuscript, we pointed out it.

C5: The scientific units need to be standardized. Is it uL or micro symbol L as both are used interchangeably? Also, the authors need to state the unit concentration of the various enzymes utilized. Merely stating the volume used is insufficient for reproducibility.

R5: Thanks for your comment. The scientific units were standardized and checked. All enzymes used in this study were purchased from the Thermo Fisher Scientific. We also inquired the technical staff of the company, and they said that the concentration of enzymes is confidential.

We hope the new version of the manuscript meets the standard of your prestigious journal, and we thank you very much for your considerations. We are looking forward to hearing from you soon. 

Best wishes,

Jin Zhao and Yao Zhang

Reviewer 2 Report

Line 6 - (J. Zhao) There is a missing dot.

Line 35 – The extra dash

Line 122 - Perhaps you should sign what the inscriptions mean. CP – coat protein, MP – maturation protein and etс. MCS, Rz, NOSt.

Line 145 - pTRV-ZjCLA or pTRV2-ZjCLA?

Line 168 – In fact, I can't see the lightened leaves very well because of the glare of the light. I don't know if it's possible to do better.

Line 179 – …maker gene… maybe marker gene?

Line 197 - pTRV2 represents the seedlings inoculated with pTRV1 + pTRV2, may be renamed, for example pTRV1+2?

Line 215 - Among nearly 40 kinds of viruses… maybe "species" or "kind of vectors". I think the second one.

Line 254 - Which Primer software was used? Please provide more details. Specify the version or it may be an online resource.

Line 267 - …to the company… You need to specify the company that sequenced or just delete " to the company".

Line 274 – …(50 mg/mL)… Is this the initial stock concentration or the final stock concentration?

Line 274 - …25 uL…  25μL

Line 277 - /min – delete. RPM - Rounds per minute

Line 291 - Is it possible to determine the concentration with a spectrophotometer? Most likely only purity by wavelength ratio 260/280, 260/230. In order to measure the concentration you need a fluorimeter and a kit, for example Equalbit RNA HS Assay Kit.

Line 292 - As far as I know, you need to specify the country and the abbreviation of the states.

Line 292 - After RNA extraction, did you treat it with DNAase or is this step included in the extraction kit?

Line 303 - 2-mL An extra dash?

Line 313 - Excuse me, but I did not understand how many plants were in the study and on what amount of data the statistics were calculated and how many replicates there were for one sample?

Line 343 and below the text - P.S.S., C.H.A. K.S. Please check the correct writing of the last names of the authors.

I did not see the following references in the text of the manuscript – 23, 43, 44, 45, 47, 48, but they are in the reference list.

In this study, the authors designed a VIGS system based on the tobacco rattle virus in Chinese jujube. Two genes ZjCLA and ZjPDS were selected for silencing. Surveillance was performed by both phenotypic traits and quantification. The authors found that the second injection increased the efficacy of silencing. The methods are described correctly and are not questionable. The results, in my opinion, are very important both in the development of molecular genetics and have practical relevance for molecular biotechnology using cultivated plants.

Author Response

Dear Reviewer,

Thank you for the comments concerning our manuscript entitled “Virus-induced gene silencing (VIGS) in Chinese jujube” ( plants-2405132 ). Those comments are very helpful for revising and improving our manuscript. We have studied your comments carefully and have made correction which we hope meet with approval. Revised portion are marked in yellow in the revised manuscript. Below you will find our detailed replies to the specific comments. Comments are pasted below in black text and our replies are in blue text. 

Reviewers’ comments are list as C.

Authors’ replies to the comments are list as R.

To Reviewer 2:

C1: Line 6 - (J. Zhao) There is a missing dot.

R1: Thanks for your careful review. We have checked and corrected the spacing error.

C2: Line 35 – The extra dash.

R2: Thanks. We corrected it.

C3: Line 122 - Perhaps you should sign what the inscriptions mean. CP – coat protein, MP – maturation protein and etс. MCS, Rz, NOSt.

R3: Thanks. According to your suggestion, we added the descriptions in the legends of Figure 2 and Figure 5.

C4: Line 145 - pTRV-ZjCLA or pTRV2-ZjCLA?

R4: Thanks. ‘pTRV-ZjCLA’ was replaced by ‘pTRV2-ZjCLA’ systematically.

C5: Line 168 - In fact, I can't see the lightened leaves very well because of the glare of the light. I don't know if it's possible to do better.

R5: Ok, Figure 4 was optimized in the revised manuscript.

C6: Line 179 - …maker gene… maybe marker gene?

R6: Yes, you are right. It should be “marker gene”, we have corrected it.

C7: Line 197 - pTRV2 represents the seedlings inoculated with pTRV1 + pTRV2, may be renamed, for example pTRV1+2?

R7: Thanks for your suggestion. ‘pTRV2’ was replaced by pTRV1+2 in the revised manuscript.

C8: Line 215 - Among nearly 40 kinds of viruses... maybe "species" or "kind of vectors". I think the second one.

R8: We have revised the description. “Among nearly 40 kinds of viruses, TRV is the most widely used virus so far to silence genes in plants.”

C9: Line 254 - Which Primer software was used? Please provide more details. Specify the version or it may be an online resource.

R9: According to your comment, we added the name of the Primer software.

C10: Line 267 - ...to the company... You need to specify the company that sequenced or just delete " to the company".

R10: Thanks, We added the name of the sequencing company.

C11: Line 274 - ...(50 mg/mL)...Is this the initial stock concentration or the final stock concentration?   

R11: Thanks, the final stock concentration was 50 mg/mL.

C12: Line 274 - ...25 uL...25 μL

R12: Thanks for your comment. ‘25 uL’ was changed by ‘25 μL’ systematically.

C13: Line 277 - /min - delete. RPM - Rounds per minute

R13: Thanks for your comment. We have removed the /min.

C14: Line 291 - Is it possible to determine the concentration with a spectrophotometer? Most likely only purity by wavelength ratio 260/280, 260/230. In order to measure the concentration you need a fluorimeter and a kit, for example Equalbit RNA HS Assay Kit.

R14: Thanks, the spectrophotometer used in this study can determine the concentration and the ratio 260/280 and 260/230. 

C15: Line 292 - As far as I know, you need to specify the country and the abbreviation of the states.

R15: Thanks, we have added the information.

C16: Line 292 - After RNA extraction, did you treat it with DNAase or is this step included in the extraction kit?

R16: Yes, DNAase was included in the extraction kit, and we added the description in Line 300-301.

C17: Line 303 - 2-mL An extra dash?

R17: Thanks for your comment. We have removed the extra dash.

C18: Line 313 - Excuse me, but I did not understand how many plants were in the study and on what amount of data the statistics were calculated and how many replicates there were for one sample?

R18: In our study, ten jujube seedlings were used as one group (one replicate), and three groups (replicates) were used in each treatment. Thus, a total of 30 plants were treated in in each treatment.

C19: Line 343 and below the text - P.S.S., C.H.A. K.S. Please check the correct writing of the last names of the authors.

R19: Thanks for your comment. We have checked and revised the correct writing of the last names of the authors.

C20: I did not see the following references in the text of the manuscript - 23, 43, 44, 45, 47, 48, but they are in the reference list.

R20: Thanks for your comment. We rechecked and revised the references

We hope the new version of the manuscript meets the standard of your prestigious journal, and we thank you very much for your considerations. We are looking forward to hearing from you soon. 

Best wishes,

Jin Zhao and Yao Zhang

Round 2

Reviewer 1 Report

In response to C1, the authors state that the unit concentration of the enzymes was proprietary. If that's the case, they should note the exact trade name/formulation of the enzymes used. I'm assuming they used FastDigest enzymes from Thermo which don't display unit concentrations. If this is the case, they should state that they used this particular formulation so that it can be reproducible by someone else purchasing the same product. While the primer efficiency of the ZjACT primers was previously characterized in Bu et al., the efficiencies of the ZjCLA and ZjPDS primers were not mentioned anywhere in the current manuscript or the cited one for the internal standards.

The English can still be improved but they are mostly minor vocabulary issues.

Could use some polishing.

Author Response

Dear Reviewers,

Thank you for the comments concerning our manuscript entitled “Virus-induced gene silencing (VIGS) in Chinese jujube” (plants-2405132). Those comments are very helpful for revising and improving our manuscript. We have studied your comments carefully and have made correction which we hope meet with approval. Revised portion are marked in yellow in the revised manuscript. Below you will find our detailed replies to the specific comments. Comments are pasted below in black text and our replies are in blue text. Reviewers’ comments are list as C. Authors’ replies to the comments are list as R. 

To Reviewer 1:

C1: In response to C1, the authors state that the unit concentration of the enzymes was proprietary. If that's the case, they should note the exact trade name/formulation of the enzymes used. I'm assuming they used FastDigest enzymes from Thermo which don't display unit concentrations. If this is the case, they should state that they used this particular formulation so that it can be reproducible by someone else purchasing the same product. While the primer efficiency of the ZjACT primers was previously  characterized in Bu et al., the efficiencies of the ZjCLA and ZjPDS primers were not mentioned anywhere in the current manuscript or the cited one for the internal standards.

R1: Thanks for your comment. We added the endonuclease information in Line 267-268.  According to your suggestion, the detection results about the efficiencies of the ZjCLA and ZjPDS primers were added in the Lines 307-311.  

We hope the new version of the manuscript meets the standard of your prestigious journal, and we thank you very much for your considerations. We are looking forward to hearing from you soon. 

Best wishes,

Jin Zhao and Yao Zhang
